# Genome-wide assessment of population structure and genetic diversity of Chinese Lou onion using specific length amplified fragment (SLAF) sequencing

Haitian Fang[1,2]*, Huiyan Liu[1,2], Ruoshuang Ma[1,2], Yuxuan Liu[1,2], Jinna Li[1,2], Xiaoyan Yu[3,4], Haoyu Zhang[1,2], Yali Yang[1,2], Guangdi Zhang[1,2,3,4]*

1 School of Agriculture, Ningxia University, Yinchuan, China, 2 Ningxia Key Laboratory for Food Microbial-Applications Technology and Safety Control, Ningxia University, Yinchuan, China, 3 Technological Innovation Center of Protected Horticulture (Ningxia University) in Ningxia, Yinchuan, China, 4 Technological Innovation center of Horticulture (Ningxia University), Ningxia Hui Autonomous Region, Yinchuan, China

* fanght@nxu.edu.cn (HF); zhangguangdi333909@sina.com (GZ)

**Data Availability Statement:** The original data has been supplied as Supporting Information. The raw sequence data reported in this paper have been deposited in the Genome Sequence Archive in BIG

## Abstract

Lou onion (*Allium fistulosum L. var. viviparum*) is an abundant source of flavonols which provides additional health benefits to diseases. Genome-wide specific length amplified fragment (SLAF) sequencing method is a rapidly developed deep sequencing technologies used for selection and identification of genetic loci or markers. This study aimed to elucidate the genetic diversity of 122 onion accessions in China using the SLAF-seq method. A set of 122 onion accessions including 107 *A.fistulosum* L. var. *viviparum* Makino, 3 *A.fistulosum* L. var. *gigantum* Makino, 3 *A.mongolicum* Regel and 9 *A.cepa* L. accessions (3 whites, 3 reds and 3 yellows) from different regions in China were enrolled. Genomic DNA was isolated from young leaves and prepared for the SLAF-seq, which generated a total of 1,387.55 M reads and 162,321 high quality SNPs (integrity >0.5 and MAF >0.05). These SNPs were used for the construction of neighbor-joining phylogenetic tree, in which 10 *A.fistulosum* L. var. *viviparum* Makino accessions from Yinchuan (Ningxia province) and Datong (Qinghai province) had close genetic relationship. The 3 *A.cepa* L. clusters (red, white and yellow) had close genetic relationship especially with the 97 *A.fistulosum* L. var. *viviparum* Makino accessions. Population structure analysis suggested entire population could be clustered into 3 groups, while principal component analysis (PCA) showed there were 4 genetic groups. We confirmed the SLAF-seq approach was effective in genetic diversity analysis in red onion accessions. The key findings would provide a reference to the Lou onion germplasm in China.

## Introduction

Lou onion (*Allium fistulosum L. var. viviparum*), also known as Chinese red onion (*Allium, Allioideae, Amaryllidaceae*), is an abundant source of flavonols with antioxidant and

Data Center, Beijing Institute of Genomics (BIG), Chinese Academy of Sciences, under submit accession number CRA002156 that are publicly accessible at https://bigd.big.ac.cn/gsa

**Funding:** This work was financially supported by the Funds of Breeding project for Agricultural Science and Technology Park in Ningxia (201406), Innovation Platform Funds of Ningxia Key Laboratory for Food Microbial-Applications Technology and Safety Control (2018DPC05026), the Key R&D Program of Ningxia (Grant No. 2017BY071), and Funds of the Western First-Class Disciplines (Horticulture) in Ningxia University (2015-2018).

**Competing interests:** The authors have declared that no competing interests exist.

antimutagenic activities [1–3]. Lou onion contains high content of quercetin derivatives [2]. Dietary supplementation of lou onion provides additional health benefits to patients with diabetes, insulin resistance, hyperglycemia and cardiovascular disease [2, 4, 5]. Lou onion is widely cultivated in China with abundant germplasm resources. Most cultivars in China could be clustered into three varietas, including *A.fistulosum* L. var. *viviparum* Makino, *A.fistulosum* L. var. *caespitosum* Makino and *A.fistulosum* L. var. *viviparum* Makino. *A.fistulosum* L. var. *viviparum* Makino has strong resistance against cold and drought and therefore is mainly cultivated in cold and arid regions (mid-east region) in China, like Gansu, Ningxia, Qinghai, Shaanxi and Sinkiang provinces. However, the germplasm of lou onion has not been systematically counted and classified till now.

Genetic resources are of great importance for germplasm identification, breeding strategy and crop improvement. Genomic tools based on molecular markers like random amplified polymorphic DNA (RAPD), inter-simple sequence repeats (ISSRs), sequence-characterized amplified region (SCAR) and amplified fragment length polymorphism (AFLP) have been effectively used to elucidate, identify and authenticate the genetic background of accessions [6–9]. What's more, the usage of molecular markers benefits to the selection and identification of genetic loci conferring different traits like color and flavonoids synthesis [10–12]. For instance, Shigyo et al reported the were different biomarker identified responsible for scale color, flower traits and abiotic stresses in onion [13]. These marker systems are economical, simple, automated and quick. However, the numbers of these marker types are insufficient to and far from saturation for large populations.

The rapidly developed deep sequencing technologies have resolved the aforementioned disadvantage of these conventional markers, like specific length amplified fragment (SLAF) sequencing method. Single nucleotide polymorphisms (SNPs) are efficient and powerful genetics and genomics studies. These burgeoning methods like SLAF-seq enable the high-throughput identification of SNPs with stability and abundance in most genomes [14, 15]. SLAF-seq has been implemented on many organisms [16, 17], including those without reference genomes [18].

This study was performed to elucidate the genetic background of 122 onion accessions in China using the genome-wide SLAF-seq method. To our knowledge, it was the first study to elucidate the genetic background of Chinese *A.fistulosum* L. var. *viviparum* Makino. This study would provide an illumination and reference for the Lou onion germplasm in China.

## Materials and methods

### Experimental plant varieties

Samples were collected in 23 different agro-climatic zones in China, with no less than 5 plant samples collected in each region in August 2015. The location of the onion accessions is shown in Fig 1. Photos and GPS position were token before sampling, then complete plants (including roots and leaves) were collected and put into the fresh-keeping bags with 4–10°C preservation. A set of 122 onion accessions, including *A.fistulosum* L. var. *viviparum* Makino (n = 107), *A.fistulosum* L. var. *gigantum* Makino (n = 3, act as outgroup), *A. mongolicum* Regel (n = 3, act as outgroup) and *A.cepa* L. (3 whites, 3 reds and 3 yellows, act as outgroup) generated from these 23 different agro-climatic zones were included and evaluated in this study. These accessions were cultivated in the agricultural science of park of arid crops in Ningxia province, China in 2016. The experimental site (105°17′-106°41′ E longitude, 36°34′-37°32′N latitude, 1283–2625 above the sea level) is characterized by an arid climate (180 mm average annual precipitation and >2000 mm average annual evaporation). Onions were placed in field fertilized with 6 m³/667 m² commercial organic fertilizer

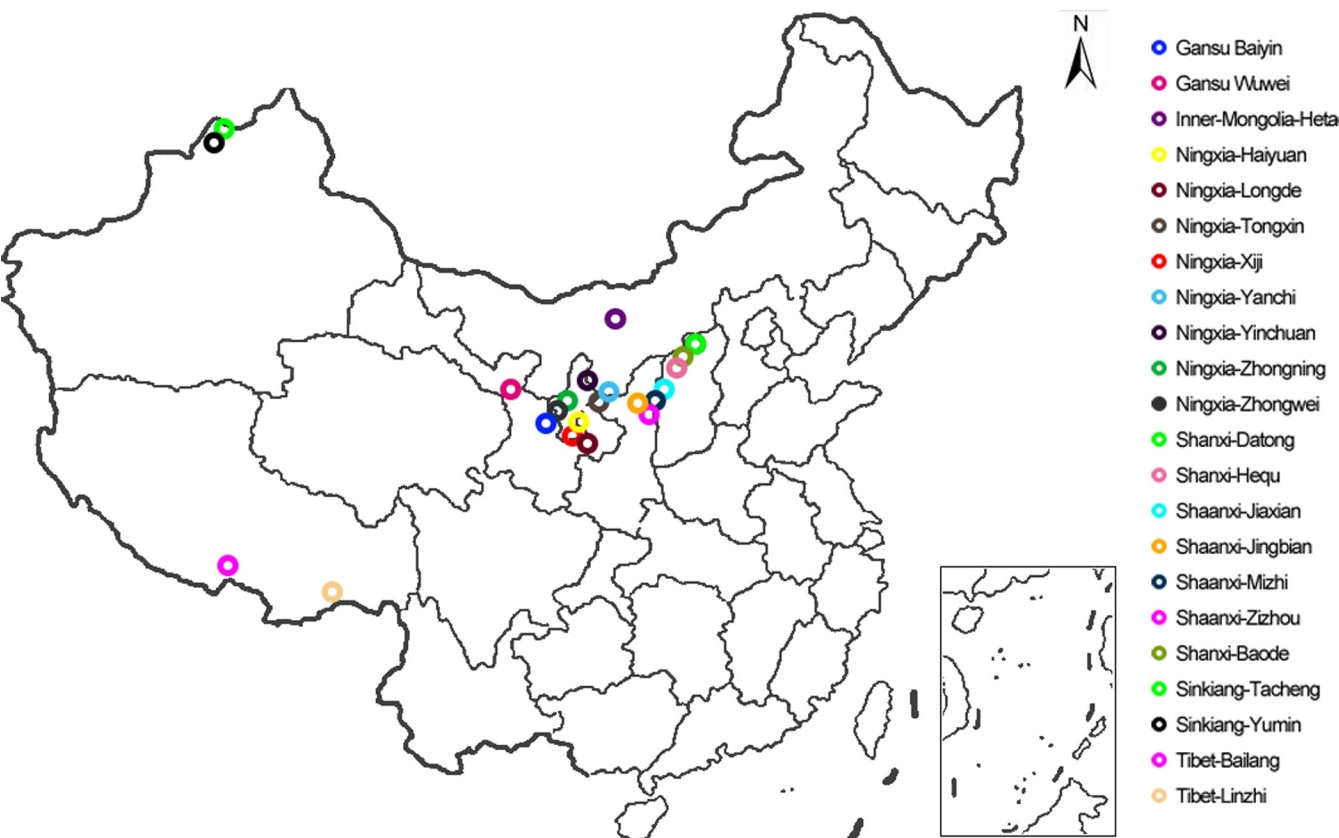

**Fig 1. Location of the Lou onion accessions evaluated in this study.** (Map sources:OSGeo China). Different colors marked different sampling locations.

(45% organic matter, 5% N+$P_2O_5$+$K_2O$, 20 million viable bacteria per gram) with micro-spray irrigation (1500 $m^3$/$hm^2$ water per year).

## Determination of the biophysical traits

The dry matter rate (B2,%) and firmness of onion bulbs (g/$cm^2$), plant weight (g), plant height (cm), the weight (g), length (cm) and diameter (cm) of pseudo stem, follower height (cm) and pseudostem index were evaluated according to previous description [19]. Three accessions of each site were included for biophysical traits evaluation and a total of 69 accessions originated from 23 sites in Tibet, Ningxia, Gansu, Shaanxi, Qinghai and Inner Mongolia. The original data were supplied in S1 Table.

## DNA extraction and high-throughput sequencing

The young leaves from 122 accessions were collected, snapped frozen in liquid nitrogen and then stored at -80˚C before DNA extraction. Fore SLAF sequencing, genomic DNA was extracted from leaf samples using the CTAB methods [20]. NanoDrop-2000 spectrophotometer (Thermo Fisher Scientific, Waltham, MA USA) was used for DNA quantification. The isolated genomic DNA was then digested with the restriction enzyme *Hea*III (New England Biolabs, Ipswich, MA, USA) and evenly distributed SLAF tags were obtained. SLAF tags (414–444 bp) were linked to Dual-index sequencing adapters and SLAF sequencing library was constructed following PCR amplification, purification, pooling and gel electrophoresis screening [21, 22]. The paired-end sequencing was performed on the Illumina HiSeq 2500 sequencing platform (Illumina, Inc.; San Diego, CA, US).

## Data processing

Raw reads were obtained and the adapter sequencings were filtered. The data quality was estimated by phred score ($Q_{30}$) and GC percentage (%). SOAP software [23] was employed for the paired-end mapping to control genome. Enzyme digestion efficiency and SLAF tags length was analyzed to evaluate the SLAF library quality. SLAF tags number and depth of each sample were evaluated. GATK [24] and SAMTOOLS [25] methods were used to call SNPs with high consistence in the sequencing population with the threshold of integrity >0.5 and minor allele frequency (MAF) > 0.05.

## Genetic evolution and statistical analyses

After data mining and SNP calling, the SNPs information was used for the analysis of genetic evolution. Phylogenetic tree was constructed using the MEGA5 software [26] with neighbor-joining algorithm (p-distance model, 1000 times bootstrap) [27]. Population structure analysis was performed using admixture software [28] based on the maximum-likelihood method and *K* ranging from 1 to 5. The cross-validation error rate of the K value was analyzed. Principal component analysis (PCA) [29] of the 122 accessions was performed using the cluster software [30].

## Statistical analysis

All data were expressed as the mean ± SD. Statistical analysis was performed using the one-way ANOVA followed with LSD test in SPSS 22.0. $P < 0.05$ was considered as significant different.

## Results

### General characteristics of *A.fistulosum L.* var. *viviparum* Makino from different locations

Table 1 shows that the accessions from the two sites at Tongxin, Ningxia, had relative higher dry matter rate and firmness, plant weight and height, pseudostem weight, length and diameter, and follow height. For instance, the pseudostem firmness of accessions from Tongxin, Ningxia was 0.85±0.02 g/cm$^2$ and 0.89±0.01 g/cm$^2$, respectively, which was insignificantly higher than 0.74±0.00% (Xiji, Ningxia), 0.71±0.01% (Haiyuan, Ningxia), 0.72±0.00% (Wuwei, Gansu) and 0.71±0.00% (Baiyin,Gansu) of accessions from Ningxia and Gansu, but were significantly higher than <0.68±0.01% from others. The dry matter rate of the accessions from Tongxin, Ningxia was 20.18±0.09% and 19.36±0.39%, respectively, which were significantly higher than < 14.06±0.24% (14.12±0.15, Wuwei, Gansu) from other accessions. The accessions from Hetao, Inner Mongolia had the highest pseudostem index (13.19 ± 1.51) compared with others (less than 7.80 ± 1.95, p < 0.05). The original data is shown in S1 Table.

### Summary of Illumina sequencing

Illumina sequencing of the 122 accessions generated 1,387.55 M reads, with a Q30 value of 92.07%, 42.79% GC percentage, and 95.01% enzyme digestion efficiency. A total of 2,168,876 SLAF tags, including 183,184 polymorphic SLAF tags (8.45%), were obtained with an average depth of 21.65-fold. The raw sequence data reported in this paper have been deposited in the Genome Sequence Archive in BIG Data Center, Beijing Institute of Genomics (BIG), Chinese Academy of Sciences, under submit accession number CRA002156 that are publicly accessible at https://bigd.big.ac.cn/gsa. The distribution of SLAF tags length is shown in Fig 2. Most of these tags were as long as 414–444 bp. SLAF-seq obtained a total of 680,825 SNPs, including

**Table 1. The comparison between different biophysical traits in *A.fistulosum* L. var. *vivip arum* Makino between different sites (n = 3).**

| Site ID | Sites | Pseudostem Firmness (g/cm²) | Dry matter rate (B2,%) | Plant weight (g) | Plant Height (cm) | Pseudostem Weight (g) | Pseudostem Length (cm) | Pseudostem Diameter (cm) | Follower height (cm)* | Pseudostem index |
|---|---|---|---|---|---|---|---|---|---|---|
| 1 | Datong, Qinghai | 0.56±0.01[e] | 9.31±0.03[cd] | 92.26±4.25[de] | 47.42±1.42[d] | 67.27±5.57[d] | 12.42±0.38[d] | 2.17±0.51[ab] | 31.92±1.59[ef] | 5.93±1.37[bc] |
| 2 | Bailang,Tibet | 0.48±0.02[f] | 6.19±0.12[f] | 80.25±1.04[f] | 49.83±1.15[c] | 55.01±1.56[e] | 13.00±0.87[cd] | 1.72±0.32[b] | 33.17±1.76[f] | 7.80±1.95[b] |
| 3 | Linzhi,Tibet | 0.57±0.02[bc] | 10.39±0.43[cd] | 88.58±1.21[e] | 56.33±3.26[abc] | 68.68±0.44[d] | 14.33±0.38[cd] | 2.23±0.78[ab] | 38.33±2.90[e] | 6.87±1.86[bc] |
| 4 | Baode,Shanxi | 0.63±0.00[d] | 9.93±0.43[cd] | 98.01±1.29[de] | 66.83±5.01[abc] | 68.01±1.29[d] | 13.17±2.25[bcd] | 2.06±0.28[ab] | 44.83±2.57[cd] | 6.37±0.25[bc] |
| 5 | Hequ,Shanxi | 0.66±0.01[bc] | 10.06±0.02[c] | 100.40±0.63[de] | 57.67±4.04[bc] | 70.40±0.63[d] | 13.33±0.63[cd] | 2.5±0.76[ab] | 39.67±0.44[e] | 5.32±0.73[c] |
| 6 | Zhongwei, Ningxia | 0.68±0.01[bc] | 10.77±0.19[cd] | 135.57±3.38[ab] | 68.92±2.67[abc] | 105.57±2.67[abc] | 14.42±3.38[bc] | 2.41±0.36[ab] | 46.67±0.29[bc] | 5.99±0.19[bc] |
| 7 | Jiaxian, Shaanxi | 0.51±0.01[e] | 8.31±0.05[de] | 107.00±2.32[c] | 61.83±1.25[bc] | 77.00±2.32[d] | 13.33±1.04[bcd] | 2.12±0.46[ab] | 41.17±1.61[dee] | 6.61±2.21[bc] |
| 8 | Mizhi, Shaanxi | 0.57±0.02[d] | 9.24±0.22[cd] | 108.80±0.80[bc] | 63.50±2.65[abc] | 78.80±0.80[d] | 14.83±1.53[bcd] | 2.29±0.20[ab] | 42.67±1.61[de] | 6.46±1.28[bc] |
| 9 | Yanchi, Ningxia | 0.64±0.01[cd] | 10.19±0.04[cd] | 141.62±1.55[ab] | 71.33±1.25[b] | 111.62±1.54[a] | 15.33±0.76[bc] | 2.35±0.50[ab] | 47.50±1.73[bcd] | 6.67±1.12[bc] |
| 10 | Zhongning, Ningxia | 0.64±0.01[c] | 10.90±0.10[cd] | 140.29±0.99[ab] | 69.67±1.26[ab] | 110.29±0.99[a] | 16.00±0.25[c] | 2.59±0.03[ab] | 44.25±0.50[cde] | 6.17±0.05[bc] |
| 11 | Tongxin-1, Ningxia | 0.85±0.02[a] | 19.36±0.39[a] | 155.96±4.69[a] | 76.00±1.80[ab] | 112.62±1.10[a] | 16.17±2.08[bc] | 2.96±0.47[ab] | 50.17±2.46[ab] | 5.17±0.76[c] |
| 12 | Hetao, Inner mongolia | 0.58±0.00[d] | 8.86±0.34[cd] | 115.57±3.38[bcd] | 76.47±1.71[ab] | 85.57±3.38[cd] | 18.40±0.64[ab] | 1.41±0.19[b] | 49.00±2.88[bc] | 13.19±1.51[a] |
| 13 | Xiji,Ningxia | 0.74±0.00[ab] | 12.05±0.08[c] | 134.87±1.22[ab] | 58.25±1.22[abc] | 104.87±3.43[b] | 15.08±0.88[bc] | 2.81±0.47[ab] | 41.72±2.88[de] | 5.51±1.33[bc] |
| 14 | Yinchuan, Ningxia | 0.63±0.01[d] | 10.16±0.03[d] | 126.99±3.51[bc] | 61.83±1.66[abc] | 96.99±3.51[bc] | 15.75±0.66[bc] | 2.84±0.59[ab] | 40.92±2.75[e] | 5.75±1.44[bc] |
| 15 | Tongxin-2, Ningxia | 0.89±0.01[a] | 20.18±0.09[a] | 158.29±2.27[a] | 78.92±0.63[a] | 118.29±2.24[a] | 17.83±1.15[bc] | 2.92±0.92[ab] | 52.83±1.89[ab] | 6.59±2.30[bc] |
| 16 | Wuwei, Gansu | 0.72±0.00[ab] | 14.12±0.15[b] | 138.22±2.04[ab] | 69.58±0.63[abc] | 108.22±2.04[a] | 15.33±0.14[c] | 2.52±0.11[ab] | 46.58±2.09[bcd] | 6.08±0.28[bc] |
| 17 | Haiyuan, Ningxia | 0.71±0.01[ab] | 11.45±0.28[c] | 134.35±1.11[ab] | 72.17±5.34[abc] | 104.35±1.11[b] | 17.25±1.09[bc] | 3.59±0.72[a] | 49.00±2.88[bc] | 4.90±0.78[c] |
| 18 | Longde, Ningxia | 0.65±0.03[bc] | 10.82±0.32[c] | 133.27±3.60[ab] | 70.50±3.20[abc] | 106.60±2.91[abc] | 23.08±11.67[abc] | 3.86±0.34[ab] | 45.92±0.52[cd] | 6.21±3.75[bc] |
| 19 | Zizhou, Shaanxi | 0.63±0.01[cd] | 8.76±0.11[cd] | 103.74±3.07[cd] | 58.00±1.80[c] | 73.74±2.75[d] | 14.83±2.75[bcd] | 2.11±0.32[ab] | 33.58±1.63[e] | 6.99±0.45[bc] |
| 20 | Tacheng, Tibet | 0.61±0.01[d] | 9.10±0.20[cd] | 120.90±1.37[bc] | 71.62±0.47[b] | 90.90±1.37[c] | 16.88±0.34[b] | 2.53±0.30[ab] | 51.67±1.77[ab] | 6.74±0.65[bc] |
| 21 | Yumin, Tibet | 0.62±0.01[d] | 10.15±0.04[c] | 122.57±1.42[bc] | 71.90±0.87[b] | 92.57±1.42[c] | 17.17±0.52[bc] | 2.49±0.22[ab] | 46.38±3.04[c] | 6.92±0.42[bc] |
| 22 | Jingbian, Shaanxi | 0.58±0.0[d] | 9.33±0.05[c] | 107.56±1.09[c] | 68.67±0.76[bc] | 77.56±1.09[d] | 15.67±1.25[bcd] | 2.40±0.72[ab] | 48.83±3.81[bc] | 6.81±1.43[bc] |
| 23 | Baiyin,Gansu | 0.71±0.00[ab] | 14.06±0.24[b] | 134.89±1.13[ab] | 70.75±2.00[ab] | 104.89±1.12[a] | 14.75±0.43[c] | 2.49±0.19[ab] | 43.63±1.10[cde] | 5.94±0.60[bc] |

*LSD test for data with homogeneity of variance. Statistically significant differences (p<0.05) between sampling sites are noted by different letters within a column. For all variables with the same letter, the difference between the means is not statistically significant. If two variables have different letters, they are significantly different.

162,321 SNPs (23.84%) with high quality in the population (integrity >0.5 and MAF >0.05). The detail information of the sequencing data of each sample is shown in S2 Table.

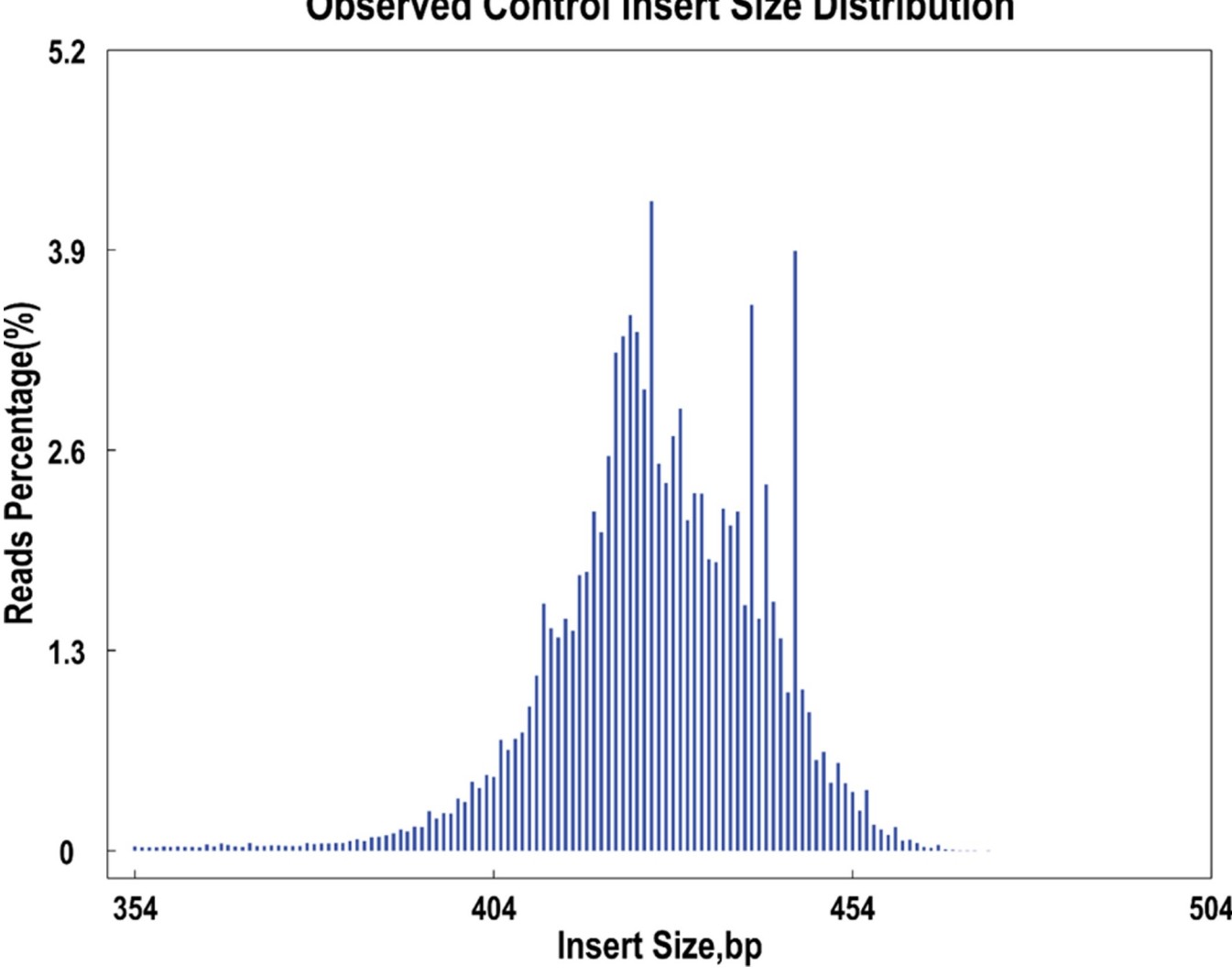

**Fig 2. The length distribution of the SLAF tags.** Abscissa represents the insertion fragment with different length. The ordinate represents the reads percentage.

## Evolutionary analysis

Based on the 162,321 high quality SNPs, we constructed the phylogenetic tree of the 122 accessions using neighbor-joining methods (Fig 3A). We found the 3 *A.fistulosum* L. var. *gigantum* Makino accessions showed close relationship with a small cluster of *A.fistulosum* L. var. *viviparum* Makino accessions (n = 10) from Yinchuan (Ningxia province) and Datong (Qinghai province); 3 *A.mongolicum* Regel accessions and 9 *A.cepa* L. accessions had long genetic distance from the other accessions, especially from the cluster of *A.fistulosum* L. var. *gigantum* Makino. Of the 107 *A.fistulosum* L. var. *viviparum* Makino accessions, 88 accessions had close genetic distance. The 3 *A.cepa* L. clusters had demarcated genetic relationship with each other and for distance from the 88 *A.fistulosum* L. var. *viviparum* Makino accessions.

Next we analyzed the genetic distances among the 107 *A.fistulosum* L. var. *viviparum* Makino accessions (Fig 3B). We observed that the 10 *A.fistulosum* L. var. *viviparum* Makino accessions from Yinchuan (Ningxia) and Datong (Qinghai) had short genetic distance between each other,

but they had long distance from the other accessions. These accessions from the two sites had no difference in dry matter rate (9.31±0.03% and 10.16±0.03%), pseudostem diameter (2.17 ±0.51 cm and 2.84±0.59 cm), follower height (31.92±1.59 cm and 40.92±2.75 cm) and pseudostem index (5.93±1.37 and 5.75±1.44, Table 1), which was consistent with the phylogenetic analysis results. Most of the samples from the same place had close genetic distance.

## Population structure

Based on the maximum-likelihood methods, population structure showed an optimum value of $K = 3$, which indicated all 122 samples in the entire population could be clustered into 3 groups (Fig 4A and 4B and S3 Table). Group 1, 2 and 3 respectively included 11, 14 and 97 accessions (Gansu, Shaanxi, Ningxia, Shanxi, Sinkiang, Tibet and Inner-Mongolia). As for group 1, *A.cepa L. accessions* were the main component; *A.fistulosum* L. var. *gigantum* Makino and *A.fistulosum* L. var. *viviparum* Makino accessions from Yinchuan, and Datong were the main components of group 2; *A. mongolicum Regel* was between group 1 and 2. As for group 3, *A.fistulosum* L. var. *viviparum* Makino accessions were the main components. The results showed that most of the samples from the same place more likely to have similar genetic contents (S3 Table).

## PCA analysis

PCA analysis separated the 122 accessions into 4 groups (Fig 5), group 1 included 9 *A.cepa* L. accessions (3 whites, 3 reds and 3 yellows); group 2 included 3 *A.fistulosum* L. var. *gigantum* Makino accessions and the 10 *A.fistulosum* L. var. *viviparum* Makino accessions originated from Yinchuan (Ningxia) and Datong (Qinghai), which have similar agronomic traits (Table 1); group 3 included 3 *A.mongolicum* Regel accessions; and the major group 4 included 97 *A.fistulosum* L. var. *viviparum* Makino accessions as indicated by the population structure analysis. These results showed that the 97 *A.fistulosum* L. var. *viviparum* Makino accessions from different sites in Gansu, Shaanxi, Ningxia, Shanxi, Sinkiang, Tibet and Inner-Mongolia had similar genetic evolution.

## Discussion

High-throughput sequencing-based genotyping technologies, like SLAF-seq, play important roles in identifying large numbers of SNPs, and other tags or molecular markers. In comparison with other sequencing methods, SLAF-seq technology reduces sequencing costs, deepens sequencing depth and ensures genotyping accuracy [31]. Restriction site associated DNA sequencing (RAD-seq) a newly developed high-throughput approach conferring comprehensive understanding of genetic evolution based on SNPs [32, 33]. In addition, RAD-seq technology is a widely used for the construction high-density genetic map [34, 35]. Lee et al. performed paired-end double digested restriction site-associated DNA sequencing (RAD-seq) and identified 1904 SNPs in 192 onion inbred lines [35]. The number of 1904 SNPs in 192 onion inbred lines by Lee et al was significantly lower than the number of 162,321 high quality SNPs and total 680,825 SNPs in 122 onion accessions in our study. In soybean, Zhao et al identified 148, 094 SNPs in 330 accessions from China and other countries [36] and Zhou et al identified 106,013 SNPs in 286 accessions from different regions in China [37]. This difference might suggest the superiority of SLAF-seq in genetic evolution analysis. In this present study, there were 680,825 SNPs identified from 122 accessions, of which 162,321 SNPs with high quality were used for the analysis of the genetic background. We found these accessions were obviously regional and varietal genetic diversity. Most of the onion plants were belong to *A.fistulosum* L. var. *viviparum* Makino. As expected, the *A.fistulosum* L. var. *gigantum* Makino, *A*.

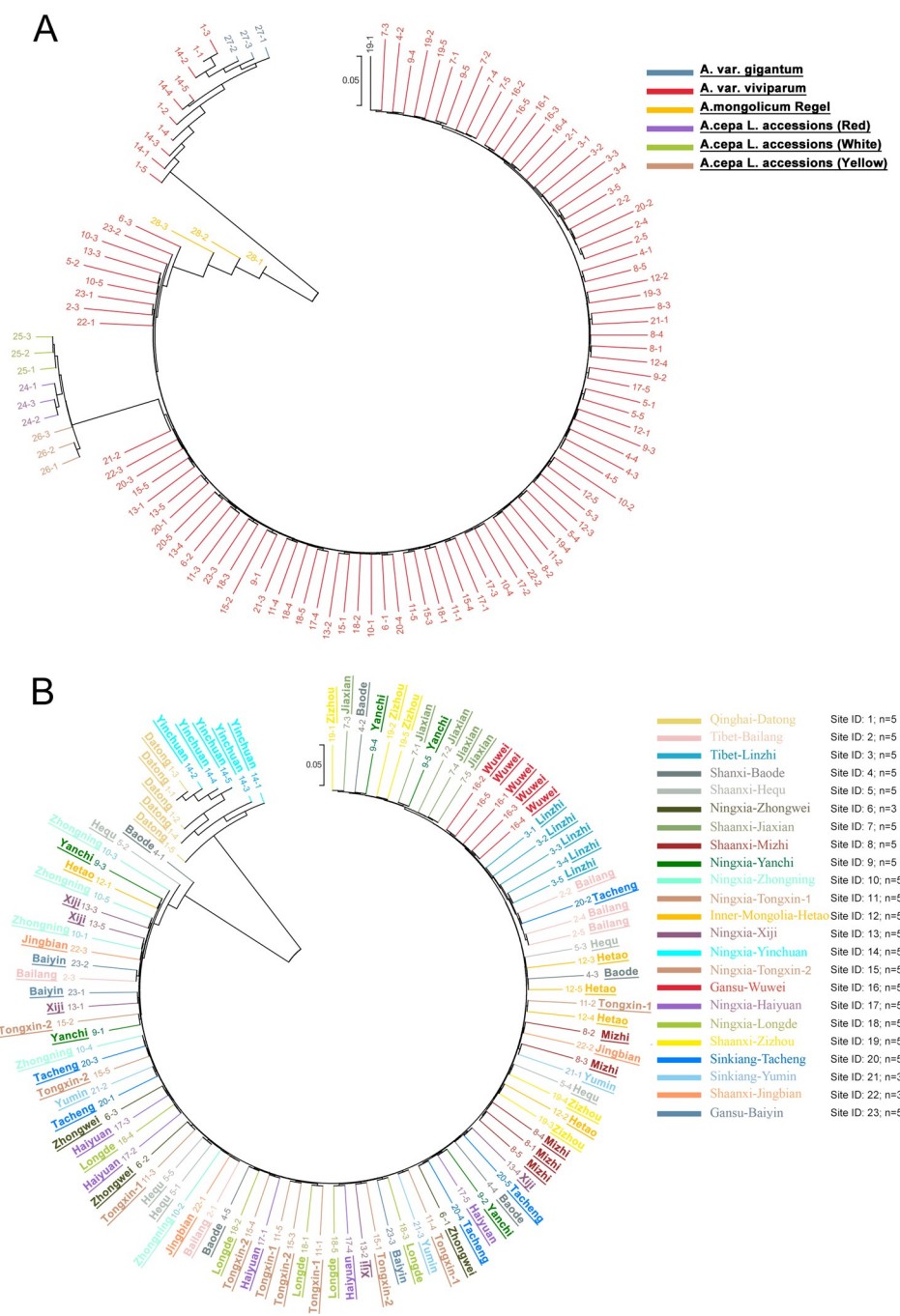

**Fig 3. The phylogenetic evolution tree of the all accessions.** A. the phylogenetic evolution tree of the 122 sequencing accessions. Different label colors represent different varieties. B, the phylogenetic evolution tree of the 107 *A.fistulosum* L. var. *viviparum* Makino accessions from 23 different sites, respectively. Different colors indicate different sampling locations. The accessions are noted by different numbers as shown in Table 1.

*mongolicum* Regel accessions and *A.cepa* L. had long genetic distance from *A.fistulosum* L. var. *viviparum* Makino, which indicated the *A.fistulosum* L. var. *viviparum* Makino had later divergence time. Interestingly, the *A.fistulosum* L. var. *viviparum* Makino from Datong and Yinchuan were clustered together with *A.fistulosum* L. var. *gigantum* Makino, which showed that

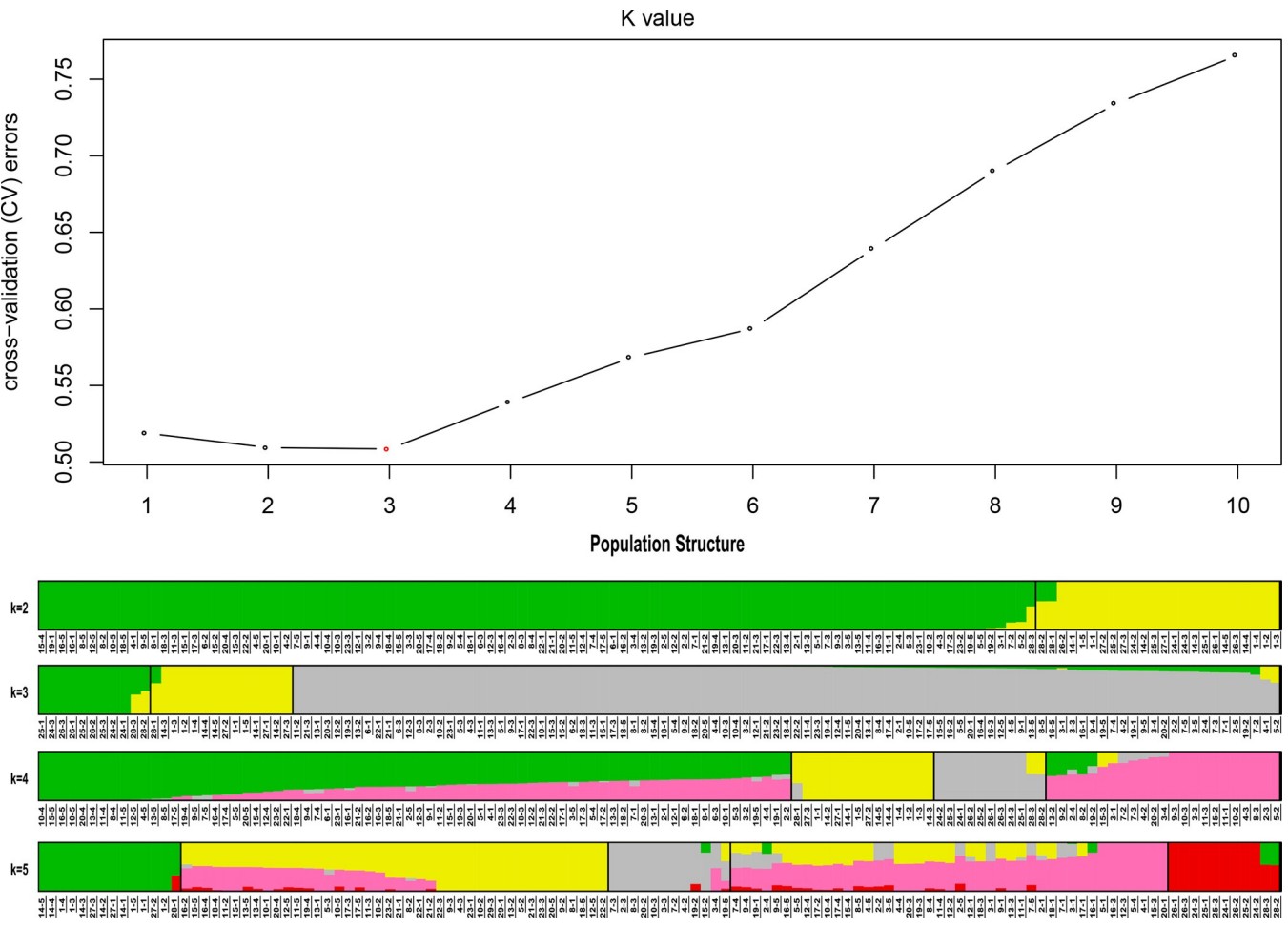

**Fig 4. Analysis results of population genetic structure.** A. The estimation of the number of groups for K value ranging from 1 to 5. The red dot indicates the optimum K value. B, the variation pattern of the 122 accessions based on the 162,321 high quality SNPs. Different colors indicated different clustered groups.

their evolutionary relationship was very close. In addition, some traits of lou onion in Datong and Yinchuan, such as pseudostem firmness and Follower height, were also similar with that of *A.fistulosum* L. var. *gigantum* Makino, which further showed the close evolutionary relationship in population between them. At the same time, lou onion in Datong and Yinchuan also showed strong abiotic stress resistance in farming practice, which was probably determined by their closer evolutionary relationship to the wild species [38–41]. In our results, The 3 *A.cepa* L. clusters (red, white and yellow) had close genetic relationship, of which the relationship was also close to *A.fistulosum* L. var. *viviparum* Makino, although their appearance is quite different, this is something we didn't expect. The appearance of *A.fistulosum* L. var. *viviparum* Makino from Bailang, Tibet and Tongxin-2, Ningxia was quite different. *A.fistulosum* L. var. *viviparum* Makino from Bailang, Tibet showed smaller pseudostem firmness, dry matter rate, plant weight, plant height and follower height, but all these indexes were higher in *A.fistulosum* L. var. *viviparum* Makino from Tongxin-2. Li et al. proposed that in non-degraded alpine meadow the maximum height of plants were highly correlated with altitude and climatic and soil variables [42]. Ma et al. found that there was a negative correlation between plant height and altitude [43]. We speculate that the differences in plant shape might be caused by different altitude and climate. After natural selection, plants form differences in appearance.

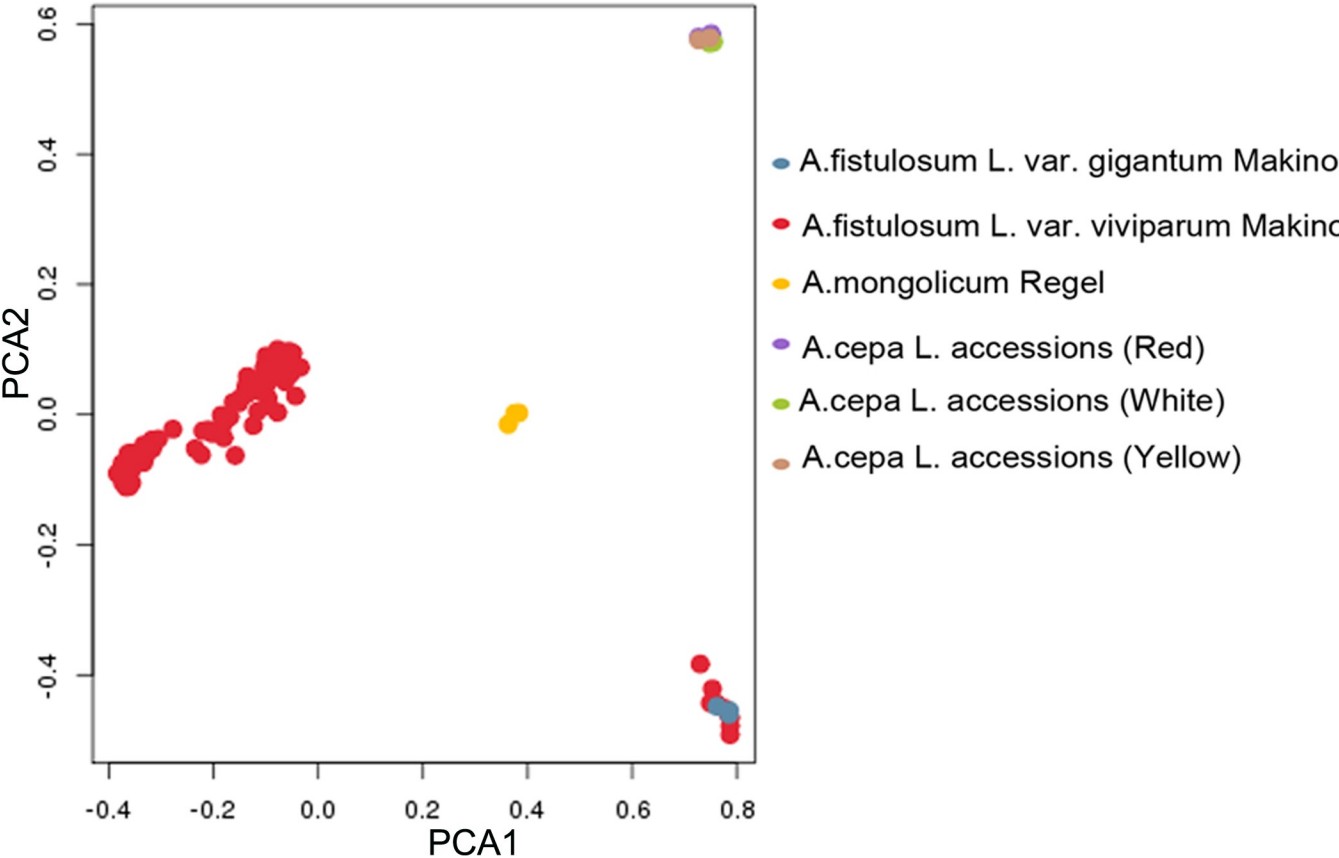

**Fig 5. Principal component analysis of the 122 accessions.** Different colors indicated different varieties.

SNP-based markers could be used for the construction of high-density genetic map, linkage map and quantitative trait loci (QTL) mapping [31, 44–46]. For instance, Chang et al reported a 4,387 polymorphism SLAF markers-based genetic map [44]. They identified a promising SNP marker closely associated with body weight might be used for the marker-assisted selection breeding [44]. In the view of genetic evolution, SLAF-seq based SNPs identification showed high efficiency [47]. Su et al. studied the phylogenetic relationships of 197 tomato accessions from China, Africa, Japan, South Korea, USA and Thailand and demonstrated the efficiency of SLAF-based SNPs on genetic diversity and core germplasm sets [47]. Our present study identified the genetic diversity in the 122 accessions of Chinese onion using SLAF methods. It was the first study to identify the genome-wide genetic diversity of Lou onion accessions in China. We obviously identified the genetic diversities among the *A.mongolicum* Regel accessions, *A.cepa* L. accessions and *A.fistulosum* L. var. *viviparum* Makino accessions. *A.fistulosum* L. var. *gigantum* Makino showed close genetic relationships to *A.fistulosum* L. var. *viviparum* Makino accessions from Yinchuan (Ningxia) and Datong (Qinghai). In addition, we found the 97 *A.fistulosum* L. var. *viviparum* Makino accessions from Gansu, Shaanxi, Ningxia, Shanxi, Sinkiang, Tibet and Inner-Mongolia of China had similar genetic history. These provide reference to our further work on breeding strategy and crop improvement.

## Conclusions

In conclusion, we confirmed the SLAF-seq approach was effective in genetic diversity analysis in Lou onion. The genetic evolution of Chinese *A.fistulosum* L. var. *viviparum* Makino

accessions was probably and scratchily described in this study. The *A.fistulosum* L. var. *viviparum* Makino accessions from Yinchuan (Ningxia) and Datong (Qinghai) showed long genetic distance from other accessions and that from other regions. However, 97 *A.fistulosum* L. var. *viviparum* Makino accessions from Gansu, Shaanxi, Ningxia, Shanxi, Sinkiang, Tibet and Inner-Mongolia of China had similar genetic history. *A.cepa* L. clusters (red, white and yellow) had close genetic relationship, of which the relationship was also close to *A.fistulosum* L. var. *viviparum* Makino, although their appearance is quite different. The differences in plant shape from Bailang, Tibet and Tongxin-2, Ningxia were quiet different, which might be caused by different altitude and climate. After natural selection, plants form differences in appearance.

## Supporting information

**S1 Table. The original data of biophysical traits in *A.fistulosum L*. var. *viviparum* Makino from different sites.**
(DOCX)

**S2 Table. Quality control of sequencing data.**
(XLSX)

**S3 Table. Analysis results of population genetic structure.**
(XLSX)

## Acknowledgments

We would like to thank, Prof. Zhanhong Ma(China Agricultural University),Prof.Jianshe Li, Prof.Yanming Gao, Prof. Wenqiang Li, Dr.Juan Shi and Dr.Peiwen Gu(Ningxia University) for samples collection and technical assistance, and We also thanked Dr. Jun Ma(Yunnan Academy of Agricultural Sciences) and colleagues of Tibet Academy of Agricultural Sciences for providing sample information in our research works.

## Author Contributions

**Conceptualization:** Haitian Fang, Guangdi Zhang.

**Data curation:** Ruoshuang Ma, Yuxuan Liu, Haoyu Zhang.

**Formal analysis:** Yali Yang.

**Investigation:** Huiyan Liu, Jinna Li, Xiaoyan Yu, Guangdi Zhang.

**Project administration:** Haitian Fang, Guangdi Zhang.

**Resources:** Haitian Fang, Xiaoyan Yu.

**Supervision:** Haitian Fang.

**Writing – original draft:** Haitian Fang.

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
