## [Decision Letter · Decision Letter 0]

16 Oct 2019

PONE-D-19-22783

Genome-wide assessment of population structure and genetic diversity of Chinese Lou onion using specific length amplified fragment (SLAF) sequencing

PLOS ONE

Dear Dr. Fang,

Thank you for submitting your manuscript to PLOS ONE. After careful consideration, we feel that it has merit but does not fully meet PLOS ONE’s publication criteria as it currently stands. Therefore, we invite you to submit a revised version of the manuscript that addresses the points raised during the review process.

We would appreciate receiving your revised manuscript by Nov 30 2019 11:59PM. To enhance the reproducibility of your results, we recommend that if applicable you deposit your laboratory protocols in protocols.io, where a protocol can be assigned its own identifier (DOI) such that it can be cited independently in the future. For instructions see: http://journals.plos.org/plosone/s/submission-guidelines#loc-laboratory-protocols

We look forward to receiving your revised manuscript.

Kind regards,

Tzen-Yuh Chiang

Academic Editor

PLOS ONE

Journal Requirements:

1. In your Data Availability statement, you have not specified where the minimal data set underlying the results described in your manuscript can be found. PLOS defines a study's minimal data set as the underlying data used to reach the conclusions drawn in the manuscript and any additional data required to replicate the reported study findings in their entirety. All PLOS journals require that the minimal data set be made fully available. For more information about our data policy, please see http://journals.plos.org/plosone/s/data-availability.

2. Please include your tables as part of your main manuscript and remove the individual files. Please note that supplementary tables (should remain/ be uploaded) as separate "supporting information" files

Reviewers' comments:

Reviewer's Responses to Questions

**Comments to the Author**

1. Is the manuscript technically sound, and do the data support the conclusions?

Reviewer #1: Partly

Reviewer #2: Partly

2. Has the statistical analysis been performed appropriately and rigorously? 

Reviewer #1: Yes

Reviewer #2: Yes

3. Have the authors made all data underlying the findings in their manuscript fully available?

Reviewer #1: No

Reviewer #2: Yes

4. Is the manuscript presented in an intelligible fashion and written in standard English?

Reviewer #1: Yes

Reviewer #2: Yes

5. Review Comments to the Author

Reviewer #1: In this manuscript, the authors described their findings about the genetic diversity of the Luo onion in China. As onion is one of the most widely grown agricultural crop, this study has some merits in understanding the genetic background of onions grown in different places in China. However, the authors only described some numbers superficially with few interpretations, and the authors did not include enough details about what they have done, making the manuscripts they wrote seems too simple. Even the figure legends are over simple, making it hard to know the points the they want to make. Thus, I suggest the authors to add more interpretation, spend some time thinking about what to illustrate in the paper and write it more careful. Below are some suggestions.

Deposit the sequenced reads to public databases such as NCBI SRA.

In the method part, the authors mentioned that “Statistical analysis was performed using the one-way ANOVA”. However, the authors did not mention where they do the test. It is in Table 1 when they were comparing the biological traits of varietas collected from different place. The authors may add more description about how they controlled the growing of those onions and how many samples they tested in each position to get the values for those traits. The numbers in table 1 seem too perfect to be true. I suggest the authors to add a supplement to include the original numbers they got. Table 1, “a-f, different letters in the same line indicate a significant different between treatments”. This description is wrong. Also, while most values have the significant group letter as superscript, some are not. What are the row “P” and the row “F”? What is “DR”? “Dunnett’3 test” should be “Dunnett’s test”. Which one you use LSD test and which “Dunnett’s test”?

Line 125, “Table 1 showed that the accessions from the two sites at Tongxi had relative higher dry matter rate and firmness, plant weight and height, pseudostem weight, length and diameter, and follow height”. The authors made the LSD test but did not draw conclusion based on the test.

Line 127, “there was not difference”, change to “there was no significant difference”. Again, I doubted the results. It is hard for a value like “13.19±1.51” not significantly different from other values in the “pseudostem index” column of Table 1. I strongly suggest the authors to include the original data for the test.

Fig. 1, use English names provinces in the figure. What are those arrows? How many samples were collected in each place and what are those samples? Which groups the samples belong to in Figure 4B? “Red star indicates the capital city of China, Beijing”. What point you want to make from this description? Help people understand the geography of China?

Fig. 3B, it is impossible to distinguish so many colors in the figure. May use abbreviations to label different groups. It’s hard to know what the labels in the figure means. Need some explanation in figure legend.

Fig. 4B, remove “k=1” part. How are the samples ordered? By varietas and by positions they were collected? Any further explanation about the observation? Are samples collected in the same places more likely to have similar genetic contents? Did you get consistent result if running admixture multiple times with different random seed numbers?

Fig. 5, I think it is better to plot 2D figures, and try to connect the results in Fig. 4B with the PCA result. Some samples from the varietas of A.fistulosum L. var. viviparum Makino are very close to A.fistulosum L. var. gigantum Makino but far from other samples from the same varietas. Any explanation?

Line 32, Principal component analysis, not “Principal components analysis”.

Line 105, add citation for SOAP software.

Line 157, “Group 1, 2 and 3 respectively included 11, 14 and 97 accessions”. So the 122 accessions includes 107 A.fistulosum L. var. viviparum Makino, 3 A.fistulosum L. var. gigantum Makino, 3 A.mongolicum Regel and 9 A.cepa L. accessions. What samples are in these three groups? Any relationship with the species?

Reviewer #2: The manuscript: “Genome-wide assessment of population structure and genetic diversity of Chinese Lou onion using specific length amplified fragment (SLAF) sequencing” is well written. The main contribution deals with the analysis of 122 A. fistulosum accessions were, as expected, regional and varietal genetic diversity were found. Also is valuable the use of the SLAF approach.

However, the biological and agronomical point of view is poorly discussed. In table 1, the authors have many data that can be discussed and they only mention that the accessions from the two sites at Tongxin, Ningxia, had relative higher dry matter rate and firmness, plant weight and height, pseudostem weight, length and diameter, and follow height. However, there was not difference in the pseudostem index. They do not relate these differences with the molecular data.

They authors also mention that SNP-based markers could be used for the construction of high-density genetic map, linkage map and quantitative trait loci (QTL) mapping, which is true as long as they have segregating populations to perform this analysis.

Some other comments:

Key words: I suggest eliminating genetic evolution, and including Allium evolution

Please review the systematics, indeed A. fistulosum is an Amaryllidaceae, but also an Allioideae (former Alliaceae). The authors should carefully review the systematics of the specie.

Instead of the title Determination of the biophysical traits, I suggest to include Agronomic Traits

I also recommend the author to review papers published in The Allium Genomes (2018) edited by Dr. Masayoshi Shigyo, published by Springer.

6. PLOS authors have the option to publish the peer review history of their article (what does this mean?). If published, this will include your full peer review and any attached files.

Reviewer #1: No

Reviewer #2: No

---

## [Author Response · Author response to Decision Letter 0]

9 Jan 2020

Dear Tzen-Yuh Chiang,

Academic Editor

PLOS ONE,

We are responding to the editor and reviewers’ comments concerning our manuscript entitled “Genome-wide assessment of population structure and genetic diversity of Chinese Lou onion using specific length amplified fragment (SLAF) sequencing” (ID: PONE-D-19-22783). 

Thank you for your letter and for the comments which are all valuable and very helpful for revising and improving our paper. We have studied the comments carefully and have made corrections as marked in the revisions and listed as follows: 

Point-by-point response to the reviewer’s comments

Journal Requirements:

1. In your Data Availability statement, you have not specified where the minimal data set underlying the results described in your manuscript can be found. PLOS defines a study's minimal data set as the underlying data used to reach the conclusions drawn in the manuscript and any additional data required to replicate the reported study findings in their entirety. All PLOS journals require that the minimal data set be made fully available. For more information about our data policy, please see http://journals.plos.org/plosone/s/data-availability.

-Response: The original data has been supplied as Supplementary Table S1. 

2. Please include your tables as part of your main manuscript and remove the individual files. Please note that supplementary tables (should remain/ be uploaded) as separate "supporting information" files

-Response: We attached the table 1 below the main text according to the comment. 

Reviewers' comments:

Reviewer's Responses to Questions

Comments to the Author

1. Is the manuscript technically sound, and do the data support the conclusions?

Reviewer #1: Partly

Reviewer #2: Partly

-Response: The results, especial Table 1, have been well displayed and discussed according to the comments from the two reviewers. 

2. Has the statistical analysis been performed appropriately and rigorously?

Reviewer #1: Yes

Reviewer #2: Yes

-Response: Thanks for your comments.

3. Have the authors made all data underlying the findings in their manuscript fully available?

The PLOS Data policyrequires authors to make all data underlying the findings described in their manuscript fully available without restriction, with rare exception (please refer to the Data Availability Statement in the manuscript PDF file). The data should be provided as part of the manuscript or its supporting information, or deposited to a public repository. For example, in addition to summary statistics, the data points behind means, medians and variance measures should be available. If there are restrictions on publicly sharing data—e.g. participant privacy or use of data from a third party—those must be specified.

Reviewer #1: No

Reviewer #2: Yes

-Response: The original data for table 1 has been supplied as Supplementary Table S1. 

4. Is the manuscript presented in an intelligible fashion and written in standard English?

Reviewer #1: Yes

Reviewer #2: Yes

-Response: Thanks for your comments. The manuscript has been improved according to other comments. However, the minor errors in grammar and writing have been checked and revised now.

5. Review Comments to the Author

Reviewer #1: 

In this manuscript, the authors described their findings about the genetic diversity of the Luo onion in China. As onion is one of the most widely grown agricultural crop, this study has some merits in understanding the genetic background of onions grown in different places in China. However, the authors only described some numbers superficially with few interpretations, and the authors did not include enough details about what they have done, making the manuscripts they wrote seems too simple. Even the figure legends are over simple, making it hard to know the points the they want to make. Thus, I suggest the authors to add more interpretation, spend some time thinking about what to illustrate in the paper and write it more careful. Below are some suggestions.

-Response: Thanks for your comments and suggestions. We have made revisions or modifications according to your kind suggestions as listed below. 

Deposit the sequenced reads to public databases such as NCBI SRA.

-Response: The raw sequence data reported in this paper have been deposited in the Genome Sequence Archive in BIG Data Center, Beijing Institute of Genomics (BIG), Chinese Academy of Sciences, under submit accession number subCRA002329 that are publicly accessible at https://bigd.big.ac.cn/gsa.

In the method part, the authors mentioned that “Statistical analysis was performed using the one-way ANOVA”. However, the authors did not mention where they do the test. 

-Response: The methods were revised to “Statistical analysis was performed using the one-way ANOVA followed with LSD test in SPSS 22.0” according to your suggestion. SPSS 22.0 was used for statistical analysis.

It is in Table 1 when they were comparing the biological traits of varietas collected from different place. The authors may add more description about how they controlled the growing of those onions and how many samples they tested in each position to get the values for those traits. 

-Response: Thank you for your comments, the sampling details were added in the newly submitted manuscript. Onions were placed in field fertilized with 6 m3/667 m2 commercial organic fertilizer (45% organic matter, 5% N+P2O5+K2O, 20 million viable bacteria per gram) with micro-spray irrigation (1500 m3/hm2 water per year). 

The numbers in table 1 seem too perfect to be true. I suggest the authors to add a supplement to include the original numbers they got.

-Response: The original data is supplied as Table S1. We promise here that all the data is true. 

Table 1, “a-f, different letters in the same line indicate a significant different between treatments”. This description is wrong. 

-Response: We changed it to “Statistically significant differences (p<0.05) between sampling sites are noted by different letters within a column”. 

Also, while most values have the significant group letter as superscript, some are not. 

-Response: Thanks for your careful review. The first two letters in the first column have been changed to superscript now. 

What are the row “P” and the row “F”? What is “DR”? “Dunnett’3 test” should be “Dunnett’s test”. Which one you use LSD test and which “Dunnett’s test”?

-Response: Sorry for the wrong writing. We perform all the statistical analyses with the LSD test.

Line 125, “Table 1 showed that the accessions from the two sites at Tongxi had relative higher dry matter rate and firmness, plant weight and height, pseudostem weight, length and diameter, and follow height”. The authors made the LSD test but did not draw conclusion based on the test.

-Response: Thanks for your comment. This part has been revised and the related sentences are “For instance, the pseudostem firmness of accessions from Tongxin, Ningxia was 0.85±0.02 g/cm2 and 0.89±0.01 g/cm2, respectively, which was insignificantly higher than 0.74±0.00% (Xiji, Ningxia), 0.71±0.01% (Haiyuan, Ningxia), 0.72±0.00% (Wuwei, Gansu) and 0.71±0.00% (Baiyin,Gansu) of accessions from Ningxia and Gansu, but were significantly higher than <0.68±0.01% from others. The dry matter rate of the accessions from Tongxin, Ningxia was 20.18±0.09% and 19.36±0.39%, respectively, which were significantly higher than < 14.06±0.24% (14.12±0.15, Wuwei,Gansu) from other accessions. The accessions from Hetao, Inner mongolia had the highest pseudostem index (13.19±1.51) compared with others (less than 7.80±1.95, p < 0.05).”

Line 127, “there was not difference”, change to “there was no significant difference”. 

-Response: This sentence has been changed as aforementioned. There are differences in the pseudostem index between sites.

Again, I doubted the results. It is hard for a value like “13.19±1.51” not significantly different from other values in the “pseudostem index” column of Table 1. I strongly suggest the authors to include the original data for the test.

-Response: Sorry for the missing analysis. We have analyzed the pseudostem index and the difference is presented in Table 1. Thanks for your careful review and the valuable correction. The original data was provided at your invitation (Supplementary Table S1).

Fig. 1, use English names provinces in the figure. What are those arrows? How many samples were collected in each place and what are those samples? Which groups the samples belong to in Figure 4B? “Red star indicates the capital city of China, Beijing”. What point you want to make from this description? Help people understand the geography of China?

-Response: Thank you for your comments. Figure 1 was reversed in the newly submitted manuscript. And the names of the sampling location were marked. As for Figure 4B, to batter explain it, we added Table S3 as supporting information.

Fig. 3B, it is impossible to distinguish so many colors in the figure. May use abbreviations to label different groups. It’s hard to know what the labels in the figure means. Need some explanation in figure legend.

-Response: This figure has been revised according to your suggestion. The explanation for the labels in the figure has been added in figure legends. 

Fig. 4B, remove “k=1” part. How are the samples ordered? By varietas and by positions they were collected? Any further explanation about the observation? Are samples collected in the same places more likely to have similar genetic contents? Did you get consistent result if running admixture multiple times with different random seed numbers?

-Response: Thank you for your comments. Samples were ordered according to the genetic diversity. Most of the samples from the same place more likely to have similar genetic contents. In the data process, we carried out the structure analysis results according to the results of linkage disequilibrium attenuation analysis. Consistent result was got with different random seed numbers.

Fig. 5, I think it is better to plot 2D figures, and try to connect the results in Fig. 4B with the PCA result. Some samples from the varietas of A.fistulosum L. var. viviparum Makino are very close to A.fistulosum L. var. gigantum Makino but far from other samples from the same varietas. Any explanation?

-Response: The 2D PCA figure was added according to your suggestion. Only three groups were identified in the 2D figure, and then the related text was revised. The correlation with Figure 4B was explanted and analyzed then.

Line 32, Principal component analysis, not “Principal components analysis”.

-Response: Done as suggested.

Line 105, add citation for SOAP software.

-Response: Reference Li et al 2009 was cited according to your suggestion. 

Line 157, “Group 1, 2 and 3 respectively included 11, 14 and 97 accessions”. So the 122 accessions includes 107 A.fistulosum L. var. viviparum Makino, 3 A.fistulosum L. var. gigantum Makino, 3 A.mongolicum Regel and 9 A.cepa L. accessions. What samples are in these three groups? Any relationship with the species?

-Response: Thank you for your suggestions, we reversed this part as suggested. 

Reviewer #2: 

The manuscript: “Genome-wide assessment of population structure and genetic diversity of Chinese Lou onion using specific length amplified fragment (SLAF) sequencing” is well written. The main contribution deals with the analysis of 122 A. fistulosum accessions were, as expected, regional and varietal genetic diversity were found. Also is valuable the use of the SLAF approach.

However, the biological and agronomical point of view is poorly discussed. In table 1, the authors have many data that can be discussed and they only mention that the accessions from the two sites at Tongxin, Ningxia, had relative higher dry matter rate and firmness, plant weight and height, pseudostem weight, length and diameter, and follow height. However, there was not difference in the pseudostem index. They do not relate these differences with the molecular data. They authors also mention that SNP-based markers could be used for the construction of high-density genetic map, linkage map and quantitative trait loci (QTL) mapping, which is true as long as they have segregating populations to perform this analysis.

-Response: Thank you very much for your kindly suggestions. In order to better express the views and conclusions of this manuscript, we revised the discussion section. The combination of different evolutionary relationships and traits were discussed. 

Some other comments:

Key words: I suggest eliminating genetic evolution, and including Allium evolution

-Response: Done as suggested. 

Please review the systematics, indeed A. fistulosum is an Amaryllidaceae, but also an Allioideae (former Alliaceae). The authors should carefully review the systematics of the specie.

-Response: Thanks for your suggestion. We changed it to “, …. also known as Chinese red onion, the Amaryllidaceae family and Allium genus (Allium, Allioideae, Amaryllidaceae”. The subfamily Allioideae is often disregarded in our mind. Thanks for your reminding. 

Instead of the title Determination of the biophysical traits, I suggest to include Agronomic Traits

-Response: Thanks for your suggestions. The inclusion of agronomic traits may increase the novelty of our paper. We determined the agronomic traits of accessions from 8 sites in Ningxia, not all the 23 sites in China. There was no consistency with the sequence analysis.

I also recommend the author to review papers published in The Allium Genomes (2018) edited by Dr. Masayoshi Shigyo, published by Springer.

-Response: Thanks for our recommendation. However, the Dr. Masayoshi Shigyo published The Allium Genomes based on the Allium Species from Middle Asia. The “Taxonomical and Ethnobotanical Aspects of Allium Species from Middle Asia” and “Classical Genetics on Gene Mapping” sections were cited. 

-Response: We tried our best to improve the manuscript and made some changes in the manuscript according to your suggestions. These changes will not influence the content and substance of the paper. And here we did not list the changes but marked in a Tracked Changes version in revised paper.

Once again, thank you very much for your comments and suggestions

Sincerely,

Dr. Tianhai Fang

---

## [Decision Letter · Decision Letter 1]

29 Jan 2020

PONE-D-19-22783R1

Genome-wide assessment of population structure and genetic diversity of Chinese Lou onion using specific length amplified fragment (SLAF) sequencing

PLOS ONE

Dear Dr. Fang,

Thank you for submitting your manuscript to PLOS ONE. After careful consideration, we feel that it has merit but does not fully meet PLOS ONE’s publication criteria as it currently stands. Therefore, we invite you to submit a revised version of the manuscript that addresses the points raised during the review process.

We would appreciate receiving your revised manuscript by Mar 14 2020 11:59PM. To enhance the reproducibility of your results, we recommend that if applicable you deposit your laboratory protocols in protocols.io, where a protocol can be assigned its own identifier (DOI) such that it can be cited independently in the future. For instructions see: http://journals.plos.org/plosone/s/submission-guidelines#loc-laboratory-protocols

Please also amend your figure as per our previous communications below.

We look forward to receiving your revised manuscript.

Kind regards,

Tzen-Yuh Chiang

Academic Editor

PLOS ONE

Journal Requirements:

Additional Editor Comments (if provided):

Reviewers' comments:

Reviewer's Responses to Questions

**Comments to the Author**

1. If the authors have adequately addressed your comments raised in a previous round of review and you feel that this manuscript is now acceptable for publication, you may indicate that here to bypass the “Comments to the Author” section, enter your conflict of interest statement in the “Confidential to Editor” section, and submit your "Accept" recommendation.

Reviewer #1: All comments have been addressed

Reviewer #2: All comments have been addressed

2. Is the manuscript technically sound, and do the data support the conclusions?

Reviewer #1: No

Reviewer #2: Yes

3. Has the statistical analysis been performed appropriately and rigorously? 

Reviewer #1: No

Reviewer #2: Yes

4. Have the authors made all data underlying the findings in their manuscript fully available?

Reviewer #1: Yes

Reviewer #2: Yes

5. Is the manuscript presented in an intelligible fashion and written in standard English?

Reviewer #1: Yes

Reviewer #2: Yes

6. Review Comments to the Author

Reviewer #1: The authors made some improvements based on the comments. But the manuscript and the response from the authors still hold some errors which are too obvious. Based on the description and the original data provided by the author, I have to draw the conclusion that the data in this manuscript is not reliable. Thus, I suggest to reject this manuscript.

The accession number is not “subCRA002329”.

Fig. 2 is already removed. So the figure legends and figure numbers in the manuscript are wrong.

The major concern is the discordant of Table 1 and Table S1. It is hard to allow such errors. In table 1, as the sample names are missing in table S1, we have to count number of samples. However, “Datong, Qinghai”, sample names are “1-1, 1-2, 1-3, 1-4, 1-5” which include 5 samples, but there are only 3 samples in table S1. “Zhongwei,Ningxia” samples are “29-1, 29-2, 29-3”, but the indexes are “16” in table S1 for that site. In fact, there are too many discordant parts in these two tables.

Reviewer #2: All the suggestions made in the previous review had been incorporated. The manuscript has improved a lot and can be published.

7. PLOS authors have the option to publish the peer review history of their article (what does this mean?). If published, this will include your full peer review and any attached files.

Reviewer #1: No

Reviewer #2: No

PONE-D-19-22783R1

Genome-wide assessment of population structure and genetic diversity of Chinese Lou onion using specific length amplified fragment (SLAF) sequencing

PLOS ONE

Dear Mr. Fang,

Thank you ever so much for your email and for confirming you will either obtain copyright permissions, or replace the figure in your manuscript. Your manuscript is with the reviewers and as soon as the academic editor has made their decision, you will be notified and then able to amend your manuscript accordingly.

Many thanks for your response and patience as your manuscript is reviewed,

Kind regards,

Vicky Stabler

PLOS ONE

On Sun, Jan 19, 2020 at 1:14 AM wrote:

Dear Dr. Vicky Stabler，

We will apply for copyright license from the original copyright holder of figure 1. If we fail to obtain the license, we will replace Figure 1 with an unprotected Figure.

SincerelyHaitian Fang

Dr.  Haitian FANG Associate professor in Food Microbiology and Fermentation EngineeringExecutive Deputy Director of Ningxia Key Laboratory for Food Microbial-Applications Technology and Safety Control.Head of Department of Food ScienceSchool of Agriculture, Ningxia UniversityYinchuan, Ningxia  750021P.R.ChinaEmail: fanght@nxu.edu.cn Phone: 86-951-2061692

--

Vicky Stabler

Editorial Office Ltdvicky.stabler@editorialoffice.co.uk=================================================================================

Running on-line journal editorial offices for societies and publishers since 2002

www.editorialoffice.co.uk

Tel: 0845 834 0370@TheEdShedConfidentiality Notice: This email and any attachments are solely for the use of the intended recipient, contain confidential and proprietary information, and may be privileged. If you are not the intended recipient (or authorized to receive messages for, or deliver them to, the intended recipient), you may not use, copy, disclose or distribute this email and any attachments. If you think you received this email in error, please notify the sender by return email or by telephone, and delete this email and any attachments from your system.

Disclaimer/Virus Notice: The views or opinions expressed in this email are those of the sender and do not necessarily represent those of Editorial Office Ltd unless otherwise specifically stated. Neither Editorial Office Ltd nor any of its agents accept any responsibility for any viruses that may be contained in this email or any attachments, and it is your responsibility to scan the email and any attachments.

In compliance with data protection regulations, you may request that we remove your personal registration details at any time. (Use the following URL: https://www.editorialmanager.com/pone/login.asp?a=r). Please contact the publication office if you have any questions.  Date: Jan 24 2020 08:00AM To: "Haitian Fang" fanght@nxu.edu.cn From: "Vicky Stabler" vicky.stabler@editorialoffice.co.uk Subject: PLOS ONE: PONE-D-19-22783R1

---

## [Author Response · Author response to Decision Letter 1]

6 Mar 2020

Dear Tzen-Yuh Chiang,

Academic Editor

PLOS ONE,

We are responding to the further comments from editor and reviewers concerning our manuscript entitled “Genome-wide assessment of population structure and genetic diversity of Chinese Lou onion using specific length amplified fragment (SLAF) sequencing” (PONE-D-19-22783R1). 

Thank you for your letter and for the comments. We have studied the comments carefully and have addressed them carefully. 

Point-by-point response to the reviewer’s comments

PONE-D-19-22783R1

Genome-wide assessment of population structure and genetic diversity of Chinese Lou onion using specific length amplified fragment (SLAF) sequencing

PLOS ONE

1. If the authors have adequately addressed your comments raised in a previous round of review and you feel that this manuscript is now acceptable for publication, you may indicate that here to bypass the “Comments to the Author” section, enter your conflict of interest statement in the “Confidential to Editor” section, and submit your "Accept" recommendation.

Reviewer #1: All comments have been addressed

Reviewer #2: All comments have been addressed

2. Is the manuscript technically sound, and do the data support the conclusions?

Reviewer #1: No

Reviewer #2: Yes

---Response: The comments of the first reviewer on our revision and data was very professional and profound. We responded the main questions on discordant parts following questions (Comment 1).

3. Has the statistical analysis been performed appropriately and rigorously? 

Reviewer #1: No

Reviewer #2: Yes

---Response: Does the first review mean the discordant parts in data? I look forward to receiving your criticism upon your specific comments, if there are any questions on the methods of statistical analysis. 

4. Have the authors made all data underlying the findings in their manuscript fully available?

Reviewer #1: Yes

Reviewer #2: Yes

5. Is the manuscript presented in an intelligible fashion and written in standard English?

Reviewer #1: Yes

Reviewer #2: Yes

6. Review Comments to the Author

Reviewer #1: The authors made some improvements based on the comments. But the manuscript and the response from the authors still hold some errors which are too obvious. Based on the description and the original data provided by the author, I have to draw the conclusion that the data in this manuscript is not reliable. Thus, I suggest to reject this manuscript.

---Response：Thanks for your comments. Your confusion might due to the inconsistency between the number of accessions used for description of biophysical traits (69 accessions originated from 23 sites) and molecular study (a total of 122 accessions originated from 23 sites and 5 out-groups). The description of biophysical traits was only performed on 69 accessions from 23 sites. It is well know that the number of samples in each group should be the same at the time of statistics. Since only three samples can be collected in some regions, we choose a sample size of 3 (n=3) for statistics. However, the number of samples does not affect the construction of evolutionary trees, so we used all the 122 samples in the phylogenetic analysis. Besides, the accessions of A.cea L. accessions (Red, 24-1, 24-2, and 24-3), A.cea L. accessions (White, 25-1, 25-2, and 25-3), A.cea L. accessions (Yellow, 26-1, 26-2, and 26-3), A.var.gigantum (27-1, 27-2, and 27-3), and A.mongolicum Regel (28-1, 28-2, and 28-3) were used as out-groups in the phylogenetic analysis, there is no need to investigate their agronomic traits.

The accession number is not “subCRA002329”.

---Response：Thanks for your comments. The submitted accession number is CRA002156, not “subCRA002329”.

Fig. 2 is already removed. So the figure legends and figure numbers in the manuscript are wrong.

---Response：Thanks for your comments. We forgot to submit Fig. 2. It has been uploaded this time.

The major concern is the discordant of Table 1 and Table S1. It is hard to allow such errors. In table 1, as the sample names are missing in table S1, we have to count number of samples. However, “Datong, Qinghai”, sample names are “1-1, 1-2, 1-3, 1-4, 1-5” which include 5 samples, but there are only 3 samples in table S1. “Zhongwei,Ningxia” samples are “29-1, 29-2, 29-3”, but the indexes are “16” in table S1 for that site. In fact, there are too many discordant parts in these two tables. 

---Response: Sorry for the inadequate description in table 1. We had added the sample number in revised MS. And we renamed the ‘29-1’, ‘29-2’, and ‘29-3’ to ‘6-1’, ‘6-2’, and ‘6-3’, respectively (There was no 6 in the original group name, but 29). So, there should be no misunderstanding. The sample number in Table S1 and S2 have been revised accordingly. In addition, the count number of accessions used for sequencing was 3-5 (a total of 122 accessions originated from 23 sites and 5 out-groups). Correspondingly, we modified the Figure 3, and we labeled the site locations and sample ID in Figure 3. For biophysical traits analysis, we used 3 individuals for each site (69 accessions originated from 23 sites, n=3). The discordant parts between the two tables have been addressed as above. We have reordered the tables (the numbers are not changed) to make it easier to read. In addition, we also marked the sample number n = 3 in Table 1. 

We thank you very much for your hard work and patience with our paper. We hope the correction and responses to the comments met your approval. 

Reviewer #2: All the suggestions made in the previous review had been incorporated. The manuscript has improved a lot and can be published.

---Response: We thank you very much for your hard work and patience with our paper.

7. PLOS authors have the option to publish the peer review history of their article (what does this mean?). If published, this will include your full peer review and any attached files.

---Response： No.

Once again, thank you very much for your comments and suggestions

Sincerely,

Dr. Fang

---

## [Decision Letter · Decision Letter 2]

1 Apr 2020

Genome-wide assessment of population structure and genetic diversity of Chinese Lou onion using specific length amplified fragment (SLAF) sequencing

PONE-D-19-22783R2

Dear Dr. Fang,

We are pleased to inform you that your manuscript has been judged scientifically suitable for publication and will be formally accepted for publication once it complies with all outstanding technical requirements.

With kind regards,

Tzen-Yuh Chiang

Academic Editor

PLOS ONE

Additional Editor Comments (optional):

Reviewers' comments:

Reviewer's Responses to Questions

**Comments to the Author**

1. If the authors have adequately addressed your comments raised in a previous round of review and you feel that this manuscript is now acceptable for publication, you may indicate that here to bypass the “Comments to the Author” section, enter your conflict of interest statement in the “Confidential to Editor” section, and submit your "Accept" recommendation.

Reviewer #2: All comments have been addressed

2. Is the manuscript technically sound, and do the data support the conclusions?

Reviewer #2: Yes

3. Has the statistical analysis been performed appropriately and rigorously? 

Reviewer #2: Yes

4. Have the authors made all data underlying the findings in their manuscript fully available?

Reviewer #2: Yes

5. Is the manuscript presented in an intelligible fashion and written in standard English?

Reviewer #2: Yes

6. Review Comments to the Author

Reviewer #2: The authors have adequately addressed the comments raised in a previous round of review and this manuscript is now acceptable for publication.

7. PLOS authors have the option to publish the peer review history of their article (what does this mean?). If published, this will include your full peer review and any attached files.

Reviewer #2: No

---

## [Editor Report · Acceptance letter]

24 Apr 2020

PONE-D-19-22783R2 

Genome-wide assessment of population structure and genetic diversity of Chinese Lou onion using specific length amplified fragment (SLAF) sequencing 

Dear Dr. Fang:

I am pleased to inform you that your manuscript has been deemed suitable for publication in PLOS ONE. Congratulations! Your manuscript is now with our production department. 

With kind regards,

on behalf of

Dr. Tzen-Yuh Chiang 

Academic Editor

PLOS ONE